# Mapping Local Climate Zones in the Urban Environment: The Optimal Combination of Data Source and Classifier

**DOI:** 10.3390/s22176407

**Published:** 2022-08-25

**Authors:** Siying Cui, Xuhong Wang, Xia Yang, Lifa Hu, Ziqi Jiang, Zihao Feng

**Affiliations:** 1College of Urban and Environmental Science, Northwest University, Xi’an 710127, China; 2Shaanxi Key Laboratory of Earth Surface System and Environmental Carrying Capacity, Northwest University, Xi’an 710127, China; 3Shaanxi Xi’an Urban Forest Ecosystem Research Station, Northwest University, Xi’an 710127, China

**Keywords:** local climate zones, GF-6 data, ShuffleNetV2, sentinel data, random forest

## Abstract

The novel concept of local climate zones (LCZs) provides a consistent classification framework for studies of the urban thermal environment. However, the development of urban climate science is severely hampered by the lack of high-resolution data to map LCZs. Using Gaofen-6 and Sentinel-1/2 as data sources, this study designed four schemes using convolutional neural network (CNN) and random forest (RF) classifiers, respectively, to demonstrate the potential of high-resolution images in LCZ mapping and evaluate the optimal combination of different data sources and classifiers. The results showed that the combination of GF-6 and CNN (S3) was considered the best LCZ classification scheme for urban areas, with OA and kappa coefficients of 85.9% and 0.842, respectively. The accuracy of urban building categories is above 80%, and the F1 score for each category is the highest, except for LCZ1 and LCZ5, where there is a small amount of confusion. The Sentinel-1/2-based RF classifier (S2) was second only to S3 and superior to the combination of GF-6 and random forest (S1), with OA and kappa coefficients of 64.4% and 0.612, respectively. The Sentinel-1/2 and CNN (S4) combination has the worst classification result, with an OA of only 39.9%. The LCZ classification map based on S3 shows that the urban building categories in Xi’an are mainly distributed within the second ring, while heavy industrial buildings have started to appear in the third ring. The urban periphery is mainly vegetated and bare land. In conclusion, CNN has the best application effect in the LCZ mapping task of high-resolution remote sensing images. In contrast, the random forest algorithm has better robustness in the band-abundant Sentinel data.

## 1. Introduction

As urbanization accelerates globally, more than half of the world’s population resides in urban areas, and this percentage is expected to gradually increase [1]. Urbanization and human activities have changed the surface energy balance, reduced the latent heat exchange between the surface and the atmosphere, and increased the anthropogenic heat release in urban areas, resulting in the urban heat island phenomenon [2,3]. This phenomenon is more pronounced in areas with rapidly increasing population densities, potentially increasing not only the threats of heat stress and heat death but also causing various environmental and climate problems [4]. For example, long-term poor ventilation in urban areas can deteriorate air quality [5]. Additionally, frequent heat waves and extreme temperature events increase urban energy consumption and affect residents’ quality of life [6,7].

The urban heat island effect is a phenomenon in which the atmospheric temperature or surface temperature in a city is higher than that in the surrounding suburbs and other nonurban environments [3,8,9]. The traditional urban–rural dichotomy is inevitably subjective in the division of urban and rural boundaries, which makes it difficult to describe the characteristics of the urban thermal environment [10,11]. Therefore, Stewart et al. proposed the Local Climate Zones (LCZs) classification system in 2012, which aims to provide an objective and standardized classification scheme for research on the urban thermal environment. LCZs are formally defined as “regions of uniform surface cover, structure, material, and human activity that span hundreds of meters to several kilometers in horizontal scale” [12,13]. In LCZ theory, the urban fabric is divided into ten urban building categories (LCZ1–10) and seven land cover categories (LCZA–G), each with consistent urban morphological characteristics related to variables such as the impervious surface fraction, average building height, surface albedo, and anthropogenic heat output [14,15,16]. LCZs utilize interclass temperature differences instead of the traditional urban–rural dichotomy, not only to compare urban structure across regions at the global scale, but also to explore the spatial differentiation patterns of different urban thermal environments in more detail. [17]. The detailed descriptive features of each category are presented in Appendix A. 

Improving the accuracy and efficiency of LCZ mapping is essential for LCZ studies. In earlier studies, LCZ mapping was mainly based on the urban morphological indicators defined by Oke et al. [13] and was performed using supervised classification and other means [18,19,20]. However, the overall accuracy and efficiency of such methods are not high, and it is challenging to perform LCZ mapping on large scales. Therefore, several community research projects have started to work on mapping LCZs at the global scale. The World Urban Database and Access Portal Tool (WUDAPT) can be used to create 100-m-resolution LCZ maps of global cities using random forest algorithms based on samples collected by users of Google Earth and using Landsat imagery as a baseline data source [21,22,23]. Since this method requires users to provide Google Earth samples, the quality of the generated maps depends on individual producers. Therefore, the average OA of LCZ maps in the WUDAPT upload library does not exceed 75%, and there is still much room for improving the classification accuracy [24,25]. Demuzere et al. proposed a sample transfer learning mechanism using the GEE platform and the existing samples in the WUDAPT library; this mechanism transferred the training samples established in one city to other cities for LCZ mapping [26]. However, this method is only suitable for cities in the same ecological zone.

Due to its robust feature learning ability, deep learning has recently been widely used in image classification and other fields [27,28,29]. Related studies have shown that convolutional neural networks outperform random forest algorithms in LCZ classification [30,31]. Based on Sentinel-1 radar and Sentinel-2 multispectral images, Zhu et al. created a benchmark sample dataset (So2Sat LCZ42) suitable for LCZ mapping [32]; they used an improved residual network to create LCZ maps (So2Sat GUL) for major urban areas around the world [33]. Zhou et al. performed LCZ mapping of three major cities in China (Beijing, Tianjin, and Wuhan) using Sentinel-1 and Sentinel-2 images as data sources [34]. Based on Sentinel-2 multispectral data, Liu et al. used the proposed LCZNet to create LCZ maps of 15 cities in China [35]. These studies showed the potential of convolutional neural networks (CNNs) in LCZ mapping; notably, CNNs not only make up for the shortcomings of traditional algorithms but also further improve the quality of LCZ mapping [35,36,37]. In current studies, Sentinel series images and Landsat series images are often used to map LCZs [38,39,40]. However, due to the substantial homogeneity of building categories in urban areas, the performance of most models with low- and medium-resolution images is poor for LCZ1-6 and LCZE. With the development of remote sensing technology, high-resolution images can enhance LCZ mapping, and using high-resolution images can provide more detailed classification results [30].

In this study, we select the main urban area of Xi’an city as the study area and use high-resolution imagery (GF-6 images with a resolution of 2 m) as input to perform LCZ mapping using a modified lightweight network, ShuffleNetV2. The potential of high-resolution remote imagery and CNNs for LCZ mapping is demonstrated by comparing traditional data source (Sentinel series images with a resolution of 10 m) and classification algorithm (random forest classifier). In addition, we compare the effect of the combination of data sources and classifiers on the classification effectiveness of LCZ and give suggestions for the optimal combination of data sources and classifiers. The objectives of this study are to (1) investigate the four schemes to demonstrate the potential of GF-6 data in mapping LCZs, and (2) compare the four schemes to determine the best LCZ combination scheme, and (3) recommend selecting combinations of data sources and classifier schemes when mapping local climate zones to alleviate the time and effort of later scholars.

## 2. Study Area and Data Sources

### 2.1. Study Area

Xi’an, the capital city of Shaanxi Province, China, and the fastest-growing inland region in Northwest China, is located from 107.4~109.5° east longitude and 33.4~34.5° north latitude (Figure 1). The rapid urbanization process and unique topographical structure (Guanzhong Basin) have led to poorly ventilated urban areas. It is difficult for aerosols and other suspended particles to diffuse to the surrounding areas, resulting in a severe urban heat island effect [41]. Xi’an frequently has extremely high-temperature weather in summer and was listed as one of the hottest ten provincial capital cities in China in 2012. As a historical and cultural city in China, to preserve the historical appearance of the city, a mix of mid-rise and low-rise buildings is mainly present within the first ring road. High-rise commercial and residential areas are densely distributed outside the first ring, with some mixed high-rise and mid-rise buildings. As the distance from the city center increases, factories, heavy industries, and low-rise residential buildings appear at the junction of the second and third rings. The city’s surrounding areas are dominated by land cover categories such as cropland and vegetation. The research in this paper mainly focuses on the main urban area of Xi’an, namely, the area indicated in the GF-6 image in Figure 1. The central urban area is the most concentrated area in Xi’an, and it includes Weiyang District, Lianhu District, Xincheng District, Beilin District, Yanta District, Baqiao District, part of Chang’an District and the Qindu District, and Weicheng District of bordering Xianyang City, with a total area of 1594 km^2^.

### 2.2. Datasets and Preprocessing

#### 2.2.1. GF-6 Data

GF-6 is a low-orbit optical remote sensing satellite in China’s High-resolution Earth Observation System (CHEOS), which was launched in 2018. The platform is equipped with a 2-m panchromatic/8-m multispectral high-resolution camera and a 16-m multispectral medium-resolution wide-format camera, which are mainly used in tasks such as precision agricultural observation and forestry resource surveying [42,43,44]. We acquired a GF-6 L2A-level image of the study area from the Natural Resources Shaanxi Province Satellite Application Technology Center; the image was previously processed by radiometric calibration, atmospheric correction, geometric correction, image fusion, and image alignment and is an RGB image covering the study area. The spatial resolution of the image was 2 m, and the imaging date was 18 September 2020.

#### 2.2.2. Sentinel-1/2 Data

Due to the high spatial and temporal resolutions and open-access availability of Sentinel series data, these data have been widely used in land cover classification [45,46], water body mapping [47], and vegetation growth monitoring [48]. We downloaded the Sentinel-1 VV + VH dual-Pol Single-Look Complex (SLC) Level 1 product covering the study area from the Copernicus Open Access Hub (https://scihub.copernicus.eu/, accessed on 1 April 2022). The date on which the image was taken was 28 August 2020. The European Space Agency’s Sentinel Application Platform (SNAP) was used to preprocess the Sentinel-1 data. The specific preprocessing technique is shown in Appendix B. The preprocessed Sentinel-1 data include alpha, entropy, and anisotropy bands. The Sentinel-2 data were acquired and processed online with the Google Earth Engine (GEE) platform; this process mainly included the following steps: (1) Sentinel-2 Level-2A images were queried and obtained based on the location of the study area, with data dated 26 August 2020, and (2) all the queried images covering the study area were mosaiced. In this study, the 60-m band data were discarded, and the 20-m band was resampled to 10 m. The Sentinel-2 image after preprocessing included ten bands, as shown in Table 1. Zhou et al. showed that the fusion of the three bands of Sentinel-1 after polarization decomposition and the ten bands of Sentinel-2 led to the creation of LCZ maps with the highest classification accuracy observed in their study [34]. Therefore, the 13 bands in Table 1 were selected in this study for layer stacking, and the stacked data were used as input features (hereafter collectively referred to as Sentinel-1/2) for LCZ mapping.

#### 2.2.3. Label Data

According to the Google Earth images of Xi’an city, the polygon samples of each LCZ category are drawn with the Google Earth Pro platform. Referring to the WUDAPT guidelines for drawing LCZ category polygons [22], we outline the homogenized polygon area for each LCZ category and ensure that the width of each polygon is greater than 200 m. Setting the reference time to a whole year enhances the classification results, so the reference time range of Google Earth images for creating polygons is set to 1 January 2020–31 December 2020. Since vegetation, water body, and bare ground are affected by the season and thus show different morphological characteristics, the polygons of natural types are mainly drawn with reference to GF-6 images. When drawing polygons, regions with unchanged building and land use types in the period should be selected, and regions that change (such as those with buildings under construction) should be avoided. The original LCZ classification system divided urban areas into ten urban building categories (LCZ1-10) and seven land cover categories (LCZA-G) [13]. In this paper, LCZs are divided into eight urban building categories (LCZ1–6, LCZ8, and LCZ10) and four land cover categories (LCZE, LCZF, LCZP, and LCZG) according to the actual situation in Xi’an city. A small amount of LCZ7 and LCZ9 were merged into LCZ3 and LCZ6, respectively, and all the natural vegetation (LCZA–D in the original classification system) was grouped into one category: LCZP.

A 96 × 96 m grid is created over the GF-6 image, and the outlined polygons are sampled within the grid. The center point of a grid falling within a polygon is labeled based on the corresponding LCZ category represented by that polygon. We labeled a total of 14,801 image pairs and boosted the number of image pairs to 17,325 using sample enhancement (up-and-down/left-and-right flipping). The GF-6 images and LCZ polygon samples collected in Google Earth images are shown in Figure 2. Additionally, one instance of each class of LCZ in Google Maps is shown on the right side of Figure 2, including but not limited to what is shown in the figure. We randomly sampled 2800 validation points within the remaining grid and marked the sampling points with types according to Google Earth images for evaluation of the accuracy of LCZ maps. Similarly, a 320 × 320 m grid was created over the Sentinel-1/2 image, and 4328 image pairs and 2800 validation points were created using the same method as described above.

## 3. Methodology

### 3.1. ShuffleNet V2 Classifier

ShuffleNetV2 [49] is an upgraded version of ShuffleNetV1 [50]. The network is designed with a structure that includes group convolution, depth-separable convolution, residual blocks, and channel shuffling. Group convolution and depth-separable convolution are utilized to reduce the computational complexity of the network (e.g., the number of FLOPs). Residual blocks are adopted to prevent gradient disappearance with increasing network depth. Channel shuffling is used to prevent information flow blockages caused by channel separation. The basic structure of ShuffleNetV2 is shown in Figure 3.

In this paper, ShuffleNetV2 is used as the basis, and the network structure is appropriately trimmed and adjusted based on the actual image size. The overall structure of the modified network is shown in Table 2. A GF-6 image is used as an example, and the input is an RGB three-channel image; the patch size is 48 × 48. The image first passes through a Conv1 convolution layer (the convolution kernel size is 3 × 3, and the step size is 1) to increase the number of channels to 32. Then, the feature map is successively passed through the Stage 2 and Stage 3 modules, and the resolution is reduced to 1/2 and 1/4 of that of the original. Each stage module consists of multiple standard shuffle blocks (Figure 3a) and a shuffle unit for down-sampling (Figure 3b). Finally, a global pooling layer, a fully connected layer, and a softmax layer are used to map the feature map to a probability distribution of 12 classes.

We comprehensively considered the spatial resolution of images, the size of each uniform LCZ block, and the CNN depth when choosing the patch size. Referring to the research of Zhu et al. [33], we finally set the patch size of the GF-6 image (spatial resolution is 2 m) to 48 × 48 and the patch size of the Sentinel-1/2 image (spatial resolution is 10 m) to 32 × 32.

The ShuffleNetV2 network was built and trained in the PyTorch framework. We allocate 80% of all labeled samples to model training and 20% to validation. The batch size was 128, the learning rate was set to 0.002, cross-entropy (*CE*, Equation (1)) was used as the loss function, Adam was used as the optimizer in the model training process [51], and Poly (${\left(1-\frac{iter}{max\;iter}\right)}^{power}$) (with power set to 2) was used for the dynamic adjustment of the learning rate [52]. The number of training iterations was set to 200.
(1)$$CE=-\frac{1}{n}\overset{n}{\underset{i=1}{\sum }}\overset{C}{\underset{j=0}{\sum }}y_{i,j}\log{\hat{y}}_{i,j}$$
where $y_{i,j}$ is the true label value, ${\hat{y}}_{i,j}$ is the model-predicted value, *C* is the number of categories, and n is the number of samples.

### 3.2. Random Forest (RF) Classifier

We chose the RF classifier as a comparative classification scheme [53,54]. RF classifiers are widely used in fields such as image classification [55,56]. A recursive binary splitting method is used to reach the final node in the tree structure. Numerous independent trees, each of which is a classifier, are generated by bootstrapping the training samples and the input variables at each tree node [57,58,59].

In this study, RF classification was performed with the GEE platform. First, the uploaded GF-6 and Sentinel-1/2 images were resampled to 96 m, and then the samples were uploaded to the GEE for training and classification. In this study, the number of random tree seeds was set to 200, and the remaining parameters were the default values. The training data for each tree accounted for 80% of the whole sample, and the remaining 20% was used for validation.

### 3.3. Classification Scheme Design

We designed four classification schemes: two RF-based classification schemes (S1 and S2) and two ShuffleNetV2-based classification schemes (S3 and S4). Different input features were used in these schemes (GF-6 and Sentinel-1/2 data). The specific combinations of data sources and classifiers are shown in Table 3. The S1 and S2 schemes are based on GF-6 (RGB bands) and Sentinel-1/2 images (13 bands), respectively, using the RF classifier for LCZ classification. In both schemes, the images are resampled to 96 m. The S3 and S4 schemes use the modified ShuffleNetV2 network for LCZ mapping, with a patch size of 48 × 48 for the S3 scheme and a patch size of 32 × 32 for the S4 scheme.

When comparing the results between the S1 and S2 schemes, the S3 and S4 schemes provide insight into the performance of images with different resolutions or different numbers of bands in the RF and CNN classifiers. When comparing the results of S1 and S3, S2 and S4 provide insight into which classifier is more suitable for high-resolution and low- and medium-resolution images. Comparing the classification results of S3 and S4 provides insight into the high-resolution image potential for improving the classification performance of the CNN classifier. In conclusion, we discuss the potential of high-resolution images in classifiers based on the fine comparison between the schemes and give the best combination of data sources and classifiers when mapping local climate zones.

### 3.4. Accuracy Assessment

All classification models were trained and validated with the training set and finally tested based on the validation set. To evaluate the accuracy of each scheme, the following metrics were selected: the overall accuracy (OA), kappa coefficient (kappa), confusion matrix, user’s accuracy (UA), producer’s accuracy (PA), F1-score (F1, Equation (2)). In particular, we added OA_urb_ (average accuracy for the urban building categories, i.e., LCZ1–6, LCZ8, and LCZ10) and OA_nat_ (average accuracy for the land cover categories, i.e., LCZE, LCZF, LCZG, and LCZP) when evaluating the LCZ classification results.
(2)$$F1=2\times \frac{\mathrm{PA}\times \mathrm{UA}}{\mathrm{PA}+\mathrm{UA}}$$

Finally, we draw the LCZ map of Xi’an city based on the map with the highest evaluation accuracy and describe the spatial layout of each type of LCZ.

## 4. Results

### 4.1. Overall Performance of the Four Schemes

Table 4 shows the results of the accuracy evaluation of the LCZ maps for the four schemes. The modified ShuffleNetV2 scheme (S3) yielded the best classification, with an OA of 85.9% and a kappa coefficient of 0.842. Additionally, OA_urb_ was 76.2%, and OA_nat_ reached 93.7%, with the best classification accuracy for the urban building and land cover categories. The classification effect for S3 was not only the best among the four schemes, but the classification accuracy was also higher than that in previous studies [15,25,60].

From Table 4, we find that the OA and kappa coefficient for the S2 scheme improved by 9.8% and 0.104 compared with those for the S1 scheme. Multiband images enhance the performance of the RF classifier more than RGB band images, mainly because multiband images provide more spectral information that can be used to better learn and analyze features. In addition, the OA and kappa coefficient of the S2 scheme were improved by 24.5% and 0.244, respectively, compared with those of the S4 scheme. From the perspective of using different classifiers, inputting well-learned features into the RF classifier yielded better classification results than did the CNN, which was consistent with the research of Athiwaratkun and Kang [61].

In classifying GF-6 images, the ShuffleNetV2 network achieved better classification performance than the RF classifier, with an OA improvement of 31.3% and a kappa coefficient improvement of 0.334 for the S3 scheme over those of the S1 scheme. Additionally, the CNN network performed significantly better than the RF in recognizing urban building categories, with a significant improvement of 39.5% in OA_urb_. Utilizing high-resolution images in the ShuffleNetV2 network enhanced classification performance, with an OA and kappa coefficient of the S3 scheme that were markedly higher than those of the S4 scheme by 46.0% and 0.474, respectively.

### 4.2. Classification Accuracy Per Class

For each LCZ class, we calculated the confusion matrices and F1 score to describe the interclass classification performance of the four schemes. Figure 4 shows the F1 score per class in the four schemes. Figure 5 shows the confusion matrices of the classification results of the four schemes.

The S1 scheme has a low F1 score for the urban building class, and most of the classes are misclassified into LCZ2, LCZ4, and LCZ6, which also means that the extraction of urban building classes using the combination of GF-6 images and RF classifier is poor. Due to the small number of GF-6 image bands, only a small number of input features can be used by the RF classifier, resulting in more confusion of urban building classes. Its F1 scores for water, vegetation, and bare land are high and meet the requirements of classification accuracy. However, the F1 score of LCZE was only 35%, and most of the misclassifications involved LCZ2 and LCZF (Figure 5a).

The F1 score of the urban building category in S2 is significantly improved compared with S1, and the F1 score of low-rise and heavy industry is above 0.6, but the F1 score of mid-rise and high-rise is below 0.6. There is a large number of misclassifications between mid-rise and high-rise buildings; while the F1 score of LCZ5 has improved, the recognition is still poor, with a large number of misclassifications to LCZ2 and LCZ4 (Figure 5b). The S2 scheme achieves the highest F1 score for water bodies, and the recognition of vegetation and bare ground is also better, but there is still much room for improvement in the recognition of LCZE.

The S3 scheme with the combination of GF-6 and CNN has the highest F1 score. The S3 scheme has the best classification of urban building classes, with the exception of LCZ1 and LCZ5, which are misclassified as LCZ4 and LCZ2, and the S3 scheme has an F1 score of nearly 0.8 for other urban building classes. LCZ1 and LCZ5 displayed significantly fewer misclassifications than other schemes, and most misclassifications were of the same type but different levels, so they were reasonable misclassifications. The F1 scores of water bodies, vegetation, and bare land in the land cover classes of the S3 scheme are above 0.95, and there are only a few misclassifications, such as a small amount of confusion between bare land and road mixed areas, and a small amount of confusion between bare land and vegetation. It is noteworthy that the F1 score of LCZE for the S3 scheme reaches 0.86, which is significantly higher than the other three schemes, which is the advantage of the combination of GF-6 and CNN. Road narrowness is not easily recognized and is easily mixed with other categories, but the combination of GF-6 images and the CNN classifier can accomplish the effective classification of mixed objects (Figure 5c).

The F1 score for the urban building category in the S4 scheme is inferior to that of the S2 and S3 schemes, but slightly higher than that of the S1 scheme. S4 has the lowest F1 score in the identification of the heavy industry and land use categories. The confusion matrix shows that LCZ8 and LCZ10 are confused with each other, LCZ1 is mostly misclassified into compact mid-rise and open high-rise, and LCZ5 is mostly classified into immediately adjacent open low-rise and heavy industrial. Overall, the S4 scheme has poor recognition of mixed features (Figure 5d). In all schemes, LCZ1 and LCZ5 exhibited different degrees of confusion, possibly because of the insufficient number of LCZ1 and LCZ5 samples in the study area, leading to poor training and classification results for these two types.

### 4.3. Comparison of LCZ Maps of the Four Schemes

Figure 6 shows LCZ maps of the four schemes. Among the four classification schemes, S3 is the most representative scheme reflecting the actual features, followed by the S2 scheme, the S1 scheme, and the S4 scheme. The urban building classes in the S3 scheme can be well distinguished, and the land cover classes, such as roads, water bodies, vegetation, and bare land, can be accurately identified. It is noteworthy that the LCZ map of the S3 scheme depicts the primary direction of the road network, and S3 is the only scheme capable of identifying LCZE (bare rock or hardened pavement) (Figure 6c). Although the classification effect of the S2 scheme is not as good as that of the S1 scheme, S2 can also effectively distinguish urban building and land cover classes. Additionally, a small amount of LCZE can be identified, with a good recognition effect for water bodies (Figure 6b).

The LCZ maps of S1 and S4 poorly show actual features in some cases. The LCZ map of S1 is scattered and messy, without considering the influence of surrounding environmental characteristics. The distinction among urban building categories is poor, with only the heavy industry and large low-rise building categories being well classified (Figure 6a). S4 is consistent with the actual situation within the city, but the division effect in the surrounding areas is poor (Figure 6d).

Figure 7 shows the LCZ classification results for three localized parts of study areas a, b, and c. Figure 7a shows a typical local area where high-rise buildings surround mid-rise buildings with nearby water bodies. Based on the LCZ classification results, the S1 and S3 schemes using high-resolution images can distinguish high-rise and mid-rise buildings well. Figure 7b shows the heavy industrial agglomeration area, and the S2 and S3 schemes are better at identifying the heavy industrial agglomeration area than the other schemes and can distinguish the mid- and low-rise residential buildings from the heavy industrial agglomeration area. However, the S2 scheme is not as effective as the S3 scheme in identifying road networks. The S3 scheme can distinguish multiple LCZ types, such as bare land, vegetation, heavy industry, and road networks. Figure 7c shows the mixed area of bare land and low-rise residential buildings. The S2 and S3 classification results are concentrated and contiguous, and they are similar to the real land structure. The S1 classification results have more misclassifications, and the S4 scheme can only identify the concentrated and contiguous areas.

### 4.4. Spatial Distribution of LCZs in the Main Part of Xi’an

The S3 scheme, which achieved the highest training and classification accuracy, was used to characterize the overall structure of the main urban area of Xi’an in 2020. As shown in Figure 7, the three-ring buffer area was used as the domain to describe the LCZ features. The buffer zone was established by selecting the Xi’an city beltway (i.e., the third ring) as the boundary and buffering 3000 and 6000 m inward to obtain the three-ring buffer zone. Overall, the center of the main urban area of Xi’an is dominated by urban buildings, and the surrounding areas are dominated by natural land use types (Figure 8a).

Based on the distance to the city center, LCZ2, LCZ4, and LCZE dominate within the first ring, occupying 35.03 km^2^, 21.20 km^2^, and 9.30 km^2^ of the area, accounting for 37.91%, 22.94%, and 10.07% of the area of the first ring, respectively. The layout of the first ring included a large area of compact mid-rise buildings, a small area of open high-rise buildings, and a regular road network. In the second ring, LCZ4 (38.63 km^2^, accounting for 25.16%) was the primary type, followed by LCZ2 (29.96 km^2^, accounting for 19.51%). A large area of LCZ10 (15.60 km^2^) appeared within the second ring, indicating that heavy industry has become prevalent, mainly in the northeast area between the second ring and the third ring. LCZ4, LCZ2, and LCZ10 are still the main types in the three rings. The natural classes in the three rings account for a large portion of the total area. Notably, LCZE, LCZF, and LCZP account for 19.10%, 7.54%, and 13.32% of the total area of the three rings, respectively (Figure 8b).

## 5. Discussion

### 5.1. Effect of the Input Bands on the RF

As shown in Table 5, the number of bands in the input image significantly impacts the performance of the RF classifier. The OA and kappa coefficient of GF-6 images were 1.2% and 0.01 higher than those of Sentinel series images when both images used the RGB band for RF classification. Therefore, the high-resolution images exhibited better classification performance when the same number of bands were input into the RF. As the number of input bands increased, the classification performance of the RF significantly improved. When the number of Sentinel-2 bands was increased to ten, the OA and kappa coefficient of the classification results significantly improved by 10.5% and 0.108, respectively. If Sentinel-2 and Sentinel-1 were used together with the RF, the OA and kappa coefficient of LCZ mapping would further improve.

The above results explain why the classification effect of the S2 scheme is better than that of the S1 scheme. The GF-6 data led to relatively poor classification performance with the RF due to the limitations associated with image acquisition (only three RGB bands in this study). Unlike GF-6 data, Sentinel-1/2 data are richer in spectral information (13 bands) and thus yield better classification results than GF-6 when used with the RF.

### 5.2. Effect of Sample Size on the CNN

The number of samples has a significant impact on the CNN classification results. The GF-6 image yielded 17,325 samples and contained polymorphic samples of each category; therefore, the CNN model had strong generalization ability and robustness. The patch size of the Sentinel image was 32 × 32, and the actual block size represented in the image size was large, resulting in a sharp decrease in the number of actual samples (4328). Therefore, CNN training and classification based on Sentinel series images were relatively poor.

A large number of training samples can improve the generalization ability of CNN models, but an unbalanced number of samples can affect the classification results. The original LCZ system included 17 categories, but some categories (such as LCZ1, LCZ3, and LCZ8) are limited or almost nonexistent (such as LCZ7 and LCZ9) in the actual surface coverage. In this study, the LCZs were reduced to 12 categories according to the actual situation, but the constructed training set still displayed a sample imbalance problem. Sample imbalance refers to different numbers of samples in each class in the classification process, which can lead to poor CNN performance [62,63]. Therefore, in this study, the oversampling method was used for data enhancement, i.e., classes with few samples were expanded by using up-and-down/left-and-right flipping. Table 6 shows that the training and classification effects were significantly better after data enhancement. The OA and kappa coefficient for GF-6 were improved by 6.5% and 0.072, respectively, while the OA and kappa coefficient for Sentinel-1/2 increased by 5.0% and 0.044, respectively, after data enhancement. In addition, data enhancement significantly improved the classification accuracy of LCZ1 and LCZ5 by 14.1% and 23.5%, respectively, compared to the accuracies with the unenhanced data.

The chosen data enhancement approach is relatively straightforward. An effective sample balancing method, such as the deep oversampling (DOS) framework [64], improved loss function [65], or adversarial minority oversampling, should be explored in future research [66].

### 5.3. Optimal Combinations of Data Sources and Classifiers

In the CNN scheme used to construct the LCZ maps, because of the spatial resolution of Sentinel images (10 m), the applicability of Sentinel series data was significantly reduced. After considering the LCZ construction scale, down-sampling depth, and the results of previous studies, the patch size of Sentinel-1/2 images was set to 32 × 32, and the actual block size was 320 × 320 m. Since a single patch may contain multiple categories, the uniformity and representativeness of some samples was poor, leading to poor model training and classification; thus, the resulting LCZ maps poorly reflected the ground features.

Using the GF-6 image (2 m) as the input significantly improved the performance of the CNN classifier, with better classification results than those obtained with the Sentinel-1/2 image. In this study, the patch size of the GF-6 image was set to 48 × 48, and the actual block size was 96 × 96 m. The LCZ classes contained in each patch were relatively simple, and the representation of the patch was better. Therefore, GF-6 images improved the training and classification performance of the CNN model. The classification effect was optimal for urban building categories and land cover categories, and urban road networks could be clearly differentiated, solving the problem that LCZE cannot be recognized or is recognized with low accuracy in traditional LCZ mapping methods.

Sentinel series images are free and easily accessible, and GF-6 images have a high resolution but are expensive. When constructing LCZ maps, the appropriate classifier should be selected according to the spatial resolution of the datasets selected. LCZ mapping based on low- and medium-resolution images, the performance of the CNN classifier is significantly weakened, so the RF classifier can be preferred. This is mainly because the CNN classifier needs to consider both the patch area and down-sampling depth. However, the RF classifier only needs to consider the size of the patch area when building the LCZ, so it requires less data resolution and obtains better classification results than the CNN in low- and medium-resolution images. In addition, the best classification results can be obtained by adding well-learned input features or an appropriate number of bands to the RF classifier in constructing LCZ maps. However, when LCZ mapping was based on high-resolution images, the CNN classifier obtained the best classification results. In this study, the GF-6 image has only RGB bands, and the RF classifier performs poorly. The high-resolution image significantly improves the classification performance of the CNN classifier and not only correctly identifies all types of LCZs but also identifies the fine roads, showing excellent classification results.

This study is still insufficient, as high-resolution images are challenging to obtain, and migration experiments involving CNN models are difficult to perform. In addition, the minimum distance between random verification points was set to 100 m, without considering the Sentinel image grid spacing in the S4 scheme, which influenced the verification accuracy of S4 to some extent.

## 6. Conclusions

In this paper, local climate zones within cities are mapped based on GF-6 images and a lightweight convolutional neural network (ShuffleNetV2). Compared with traditional mapping methods, this scheme effectively improves the classification accuracy of LCZ, fully illustrates the potential of high-resolution images in LCZ mapping and is considered the best scheme for mapping local climate zones in urban areas. The scheme is expected to help researchers map urban local climate zones using GF-6 images, and its mapping results can also provide useful information for urban planning authorities to regulate the urban thermal environment at a more detailed level and reduce the urban heat island effect.

We provide a detailed comparison of the four designed schemes and discuss the reasons for the differences in classification accuracy. Finally, we give the best recommendations for the combination of data sources and classifiers in mapping LCZs. The results show that the low- and medium-resolution images are poorly adapted to the CNN classifier when mapping LCZs, so the RF classifier can be given priority when mapping. In contrast, the combination of the high-resolution images and the CNN classifier can map the best local climate zones in urban areas. This provides reasonable suggestions for urban thermal environment researchers and reduces the time and labor cost for researchers to select data sources and classifiers in mapping LCZs.

## Figures and Tables

**Figure 1 sensors-22-06407-f001:**
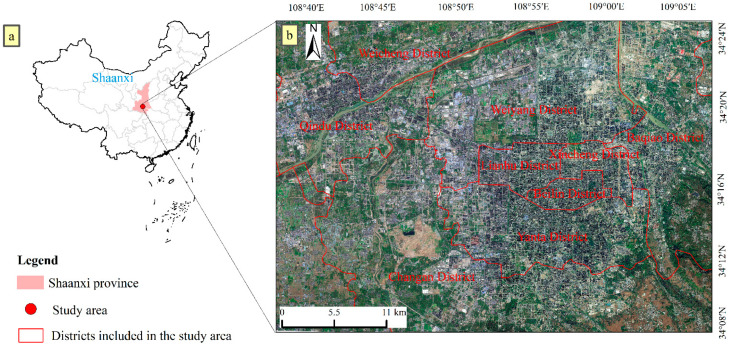
Location of Shaanxi Province and the study area in China (**a**); schematic view of the main city of Xi’an from a GF-6 image (**b**).

**Figure 2 sensors-22-06407-f002:**
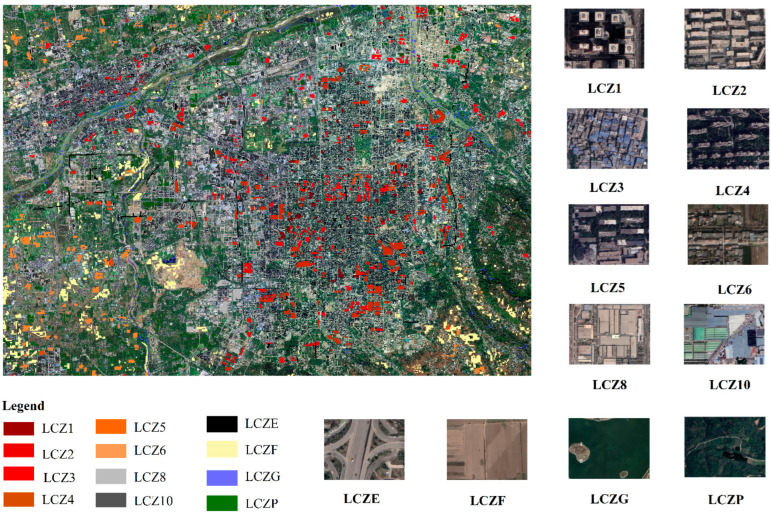
GF-6 images and LCZ labels outlined in Google Earth; schematic diagrams of Google Earth images for each category of LCZ are shown on the right, including but not limited to the above examples.

**Figure 3 sensors-22-06407-f003:**
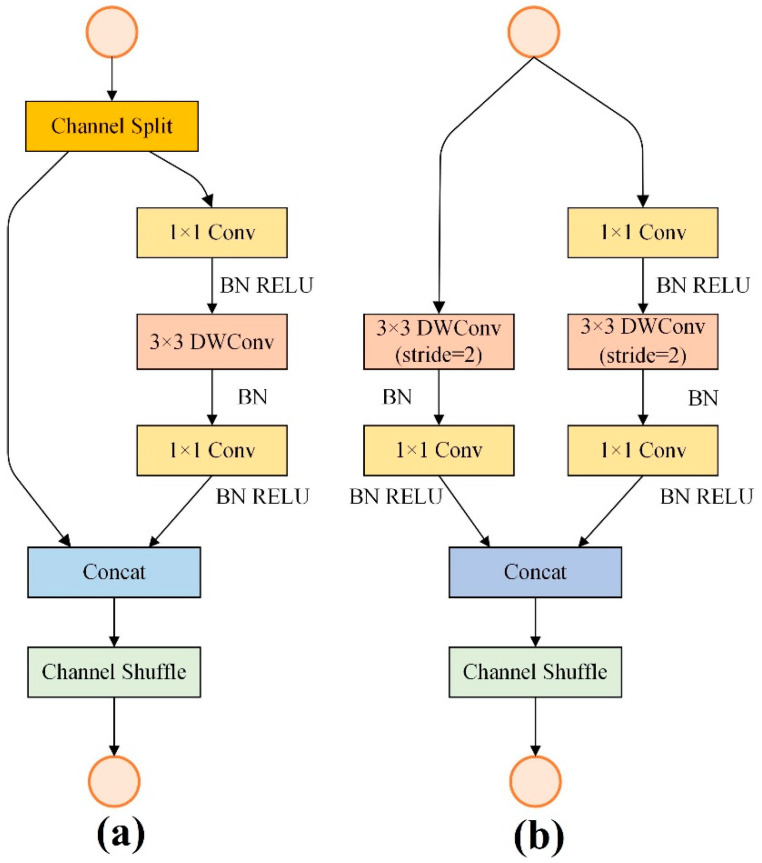
Standard Shuffle unit (**a**); Shuffle unit for down-sampling (**b**) Reprinted/adapted with permission from Ref. [49]. 2018, ©Springer Nature.

**Figure 4 sensors-22-06407-f004:**
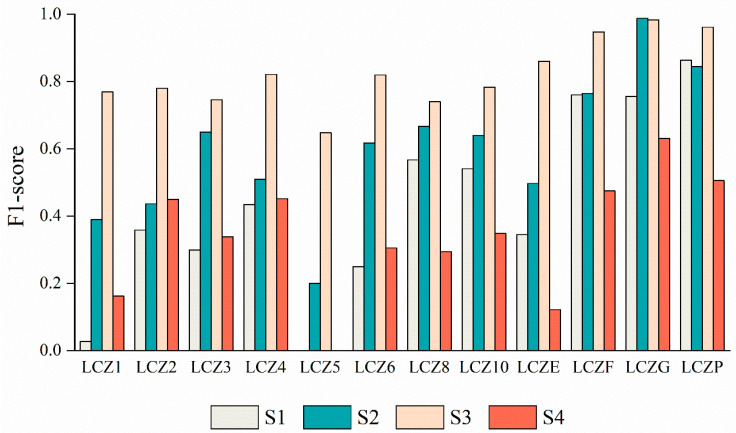
F1-score per class of LCZ in four schemes.

**Figure 5 sensors-22-06407-f005:**
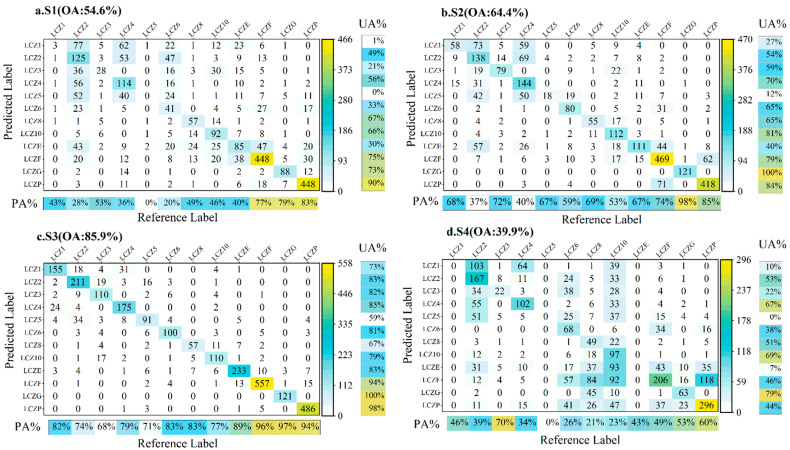
Confusion matrix of the S1 scheme (**a**); confusion matrix of the S2 scheme (**b**); confusion matrix of the S3 scheme (**c**); confusion matrix of the S4 scheme (**d**).

**Figure 6 sensors-22-06407-f006:**
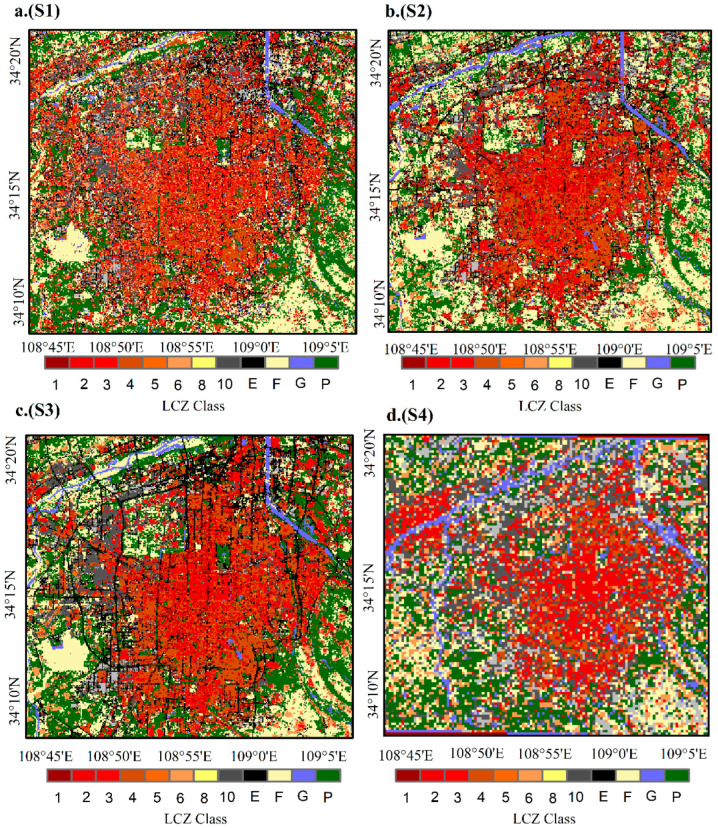
LCZ map of scheme S1 (**a**); LCZ map of scheme S2 (**b**); LCZ map of scheme S3 (**c**); LCZ map of scheme S4 (**d**).

**Figure 7 sensors-22-06407-f007:**
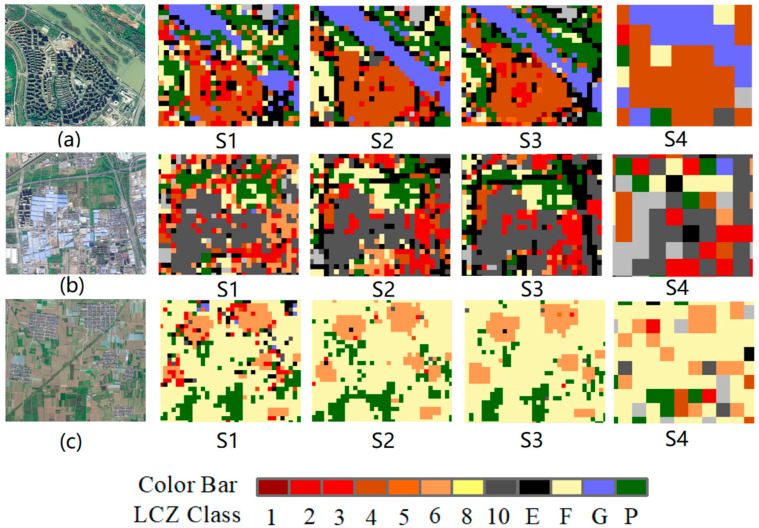
LCZ classification results for three localized regions of (**a**–**c**). The GF-6 image is shown on the left, and the LCZ results of the four schemes are shown on the right.

**Figure 8 sensors-22-06407-f008:**
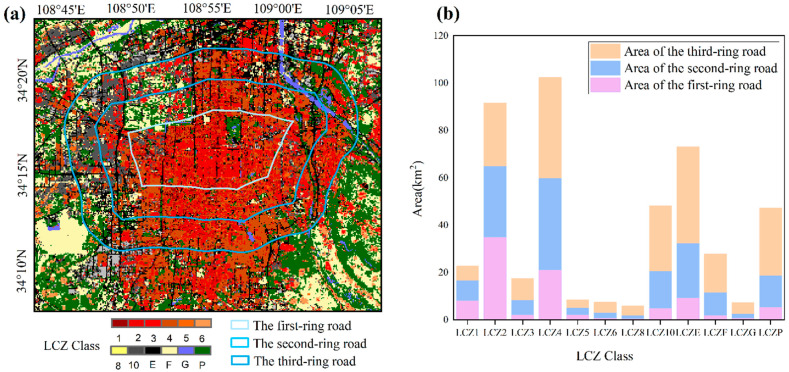
Distribution of LCZs within the three rings (**a**); histogram of the area of LCZs in the third ring (**b**).

**Table 1 sensors-22-06407-t001:** The bands used in GF-6, Sentinel-1, and Sentinel-2.

No.	GF-6	Sentinel-1	Sentinel-2
1	Red	Alpha	B2
2	Green	Entropy	B3
3	Blue	Anisotropy	B4
4	\	\	B5
5	\	\	B6
6	\	\	B7
7	\	\	B8
8	\	\	B8a
9	\	\	B11
10	\	\	B12

**Table 2 sensors-22-06407-t002:** Modified ShuffleNetV2 network.

Layer	Output Size	Ksize	Stride	Repeat	Output Channels
Image	48 × 48				3
Conv1	48 × 48	3 × 3	1	1	32
Stage 2	24 × 24		2	1	64
	24 × 24		1	1	
Stage 3	12 × 12		2	1	128
	12 × 12		1	3	
GlobalPool	1 × 1				
FC					12
Softmax					12

**Table 3 sensors-22-06407-t003:** Comparative scheme design.

Scheme	Classifier	Input Data	Feature Types (Spatial Resolution)
S1	RF	GF-6	96 m
S2	RF	Sentinel-1/2	96 m
S3	ShuffleNet V2	GF-6	Size 48 × 48 (2 m)
S4	ShuffleNet V2	Sentinel-1/2	Size 32 × 32 (10 m)

**Table 4 sensors-22-06407-t004:** Accuracy of the four schemes.

Scheme	OA	Kappa	OA_urb_	OA_nat_
S1	54.6%	0.508	36.7%	67.2%
S2	64.4%	0.612	54.0%	75.7%
S3	85.9%	0.842	76.2%	93.7%
S4	39.9%	0.368	38.7%	44.0%

**Table 5 sensors-22-06407-t005:** Influence of the input bands on the RF classification results.

Data	Classifier	Input Bands	OA	Kappa
GF	RF	RGB	54.6%	0.508
Sentinel-2	RF	RGB	53.4%	0.498
Sentinel-2	RF	10 bands	63.9%	0.606
Sentinel-1/2	RF	13 bands	64.4%	0.612

**Table 6 sensors-22-06407-t006:** Comparison of CNN classification results before and after data enhancement.

Data	Sample	OA	Kappa
GF-6	14,000	79.4%	0.770
GF-6	17,325	85.9%	0.842
Sentinel-1/2	3805	34.9%	0.324
Sentinel-1/2	4750	39.9%	0.368

## Data Availability

The data presented in this study are available on request from the corresponding author.

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
