# Peer review of "Mapping Local Climate Zones in the Urban Environment: The Optimal Combination of Data Source and Classifier"

_sensors, 2022, doi:10.3390/s22176407_

Round 1

Reviewer 1 Report

See attached files 

Author Response

Thank you very much for your positive comments and constructive suggestions on our manuscript. We appreciate your efforts in reviewing our manuscript. We have uploaded the response to the attachment, please check it out.

Reviewer 2 Report

I like the authors thorough presentation of methods and data.  I feel that you are still combining two variables here.  When you go between the S3 and S4, you are changing both number of bands (3 to 13) and patch size.  Why did you not also do 13 bands and 48 x 48 patch size?  Would that not have provided the best expected results?  You reference the work of Zhu et. al. and say that you "comprehensively considered" the spatial resolution of the images, but perhaps you could expand on the thought process as that seems to have a direct impact on the results of your experiment.  As it is, you are putting the S4 method at a spatial resolution disadvantage that doesn't have a non-CNN classifier to compare to it with similar resolution.  

Author Response

Thank you very much for your positive comments and constructive suggestions on our manuscript. We appreciate your efforts in reviewing our manuscript. We have uploaded the response to the attachment. Please check it out.

This manuscript is a resubmission of an earlier submission. The following is a list of the peer review reports and author responses from that submission.